# TRANSFER LEARNING TO LEARN WITH MULTITASK NEURAL MODEL SEARCH

## ABSTRACT

Deep learning models require extensive architecture design exploration and hyperparameter optimization to perform well on a given task. The exploration of the model design space is often made by a human expert, and optimized using a combination of grid search and search heuristics over a large space of possible choices. Neural Architecture Search (NAS) is a Reinforcement Learning approach that has been proposed to automate architecture design. NAS has been successfully applied to generate Neural Networks that rival the best human-designed architectures. However, NAS requires sampling, constructing, and training hundreds to thousands of models to achieve well-performing architectures. This procedure needs to be executed from scratch for each new task. The application of NAS to a wide set of tasks currently lacks a way to transfer generalizable knowledge across tasks.

In this paper, we present the Multitask Neural Model Search (MNMS) controller. Our goal is to learn a generalizable framework that can condition model construction on successful model searches for previously seen tasks, thus significantly speeding up the search for new tasks. We demonstrate that MNMS can conduct an automated architecture search for multiple tasks simultaneously while still learning well-performing, specialized models for each task. We then show that pre-trained MNMS controllers can transfer learning to new tasks. By leveraging knowledge from previous searches, we find that pre-trained MNMS models start from a better location in the search space and reduce search time on unseen tasks, while still discovering models that outperform published human-designed models.

## 1 INTRODUCTION

Designing deep learning models that work well for a task requires an extensive process of iterative architecture engineering and tuning. These design decisions are largely made by human experts guided by a combination of intuition, grid search, and search heuristics.

Meta-learning aims to automate model design by using machine learning to discover good architecture and hyperparameter choices. Recent advances in meta-learning using Reinforcement Learning (RL) have made promising strides towards accelerating or even eliminating the manual parameter search. For example, Neural Architecture Search (NAS) has successfully discovered novel network architectures that rival or surpass the best human-designed architectures on challenging benchmark image recognition tasks (Zoph & Le, 2017; Zoph et al., 2017). However, naively applying reinforcement learning to each new task for automated model construction requires sampling, constructing, and training hundreds to thousands of networks to relearn how to generate models from scratch. Human experts, on the other hand, can design and tune networks based on knowledge about underlying dependencies in the search space and experience with prior tasks. We therefore aim to automatically learn and leverage the same information.

In this paper, we present Multitask Neural Model Search (MNMS), an automated model construction framework that finds the best performing models in the search space for multiple tasks simultaneously. We then show that a MNMS framework that has been pre-trained on previous tasks can construct the best performing model for entirely new tasks in significantly less time.

## 2 RELATED WORK

The Neural Architecture Search (NAS) method was introduced in (Zoph & Le, 2017), where it was applied to construct Convolutional Neural Networks (CNNs) for the CIFAR-10 task and Recurrent Neural Networks (RNNs) for the Penn Treebank tasks. Later work by the same authors attempted to address the computational cost of using Neural Architecture Search for more challenging tasks (Zoph et al., 2017). To engineer a convolutional architecture for ImageNet classification, this paper demonstrated that it was possible to train the NAS controller on the simpler, proxy CIFAR-10 task and then transfer the architecture to ImageNet classification by stacking it. However, this work did not attempt to transfer learn the NAS controller itself across multiple tasks, relying instead on the human expert intuition that additional network depth was necessary for the more challenging classification task. Additionally, the final generated architectures required additional tuning, to choose hyperparameters such as the learning rate, before evaluation on the test set.

The complexity of model engineering in machine learning is widely recognized. Optimization methods have been proposed, ranging from random search over the space of possible architectures (Bergstra & Bengio, 2012) to parameter modeling (Bergstra et al., 2013). Recent publications apply RL to automate architecture generation. These include MetaQNN, a Q-learning algorithm that sequentially chooses CNN layers (Baker et al., 2016). MetaQNN uses an aggressive exploration to reduce search time, though it can cause the resulting architectures to underperform. Cai et al. (2017) also propose an RL agent that transforms existing architectures incrementally to avoid generating entire networks from scratch. More recently, an emerging body of modern neuro-evolution research has adapted genetic algorithms as an alternate optimization method for these complex searches (Conti et al., 2017).

Our work also draws on prior research in transfer learning and simultaneous multitask training. Transfer learning has been shown to achieve excellent results as an initialization method for deep networks, including for models trained using RL (Yosinski et al., 2014; Sharif Razavian et al., 2014; Zhan & Taylor, 2015). Simultaneous multitask training can also facilitate learning between tasks with a common structure, though effectively retaining knowledge across tasks is still an active area of research Kirkpatrick et al. (2017); Teh et al. (2017).

## 3 METHODS

### 3.1 NEURAL ARCHITECTURE SEARCH OVERVIEW

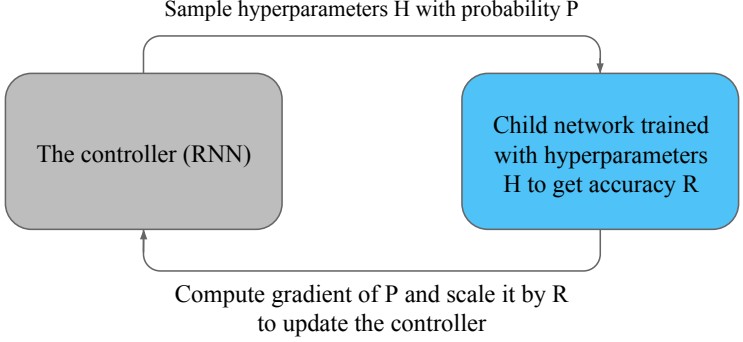

Figure 1: The base Neural Architecture Search framework.

Neural Architecture Search uses an RNN to generate model designs that maximize expected performance on a given task (Figure 1) (Zoph & Le, 2017). Specifically, an RNN controller iteratively samples architectures as a sequence of actions. Every action is a discretized design choice, such as CNN filter heights, widths, and strides. Child networks are then constructed with these architectures and trained to convergence. The performance metric of the child network is used as a reward to update the controller through a policy gradient algorithm. The controller learns a distribution over

the architecture search space that is updated to increase the probability of the best performing architectures, allowing it to sample better architectures over time. The original Neural Architecture Search framework sampled neural network models over a search space of strictly architectural parameters, by generating descriptions of each layer in the network at a time. This framework was used to successfully specify a convolutional neural network architecture for image classification, and a recurrent network cell for language tasks (Zoph & Le, 2017). Later work has shown that this framework can be extended to automatically search over other model design parameters and domains, such as update rules for network optimizers (Bello et al., 2017).

### 3.2 Simultaneous Multitask Training in Neural Architecture Search

In this section, we describe the Multitask Neural Model Search (MNMS) controller, which allows simultaneous model search over multiple different tasks. Many deep learning models require the same common design decisions, such as choice of network depth, learning rate, and number of training iterations; using a generally defined search space of widely applicable architecture and hyperparameter choices, the controller can therefore engineer a wide range of models applicable to many common machine learning tasks. Multitask training over this space can then allow the controller to learn more broadly applicable relationships between search space actions, by leveraging shared behavior across tasks.

We implement a controller capable of simultaneous multitask training through three key modifications:

1. *Learned task representation and task conditioning.*

   The MNMS controller can be trained synchronously on a set of N tasks. The controller learns to build differentiated architectures for each task. This is achieved by sampling a task uniformly at the beginning of each controller training iteration. The task is then mapped to a unique embedding vector. The tasks embeddings are randomly initialized, and are trained jointly with the controller. The task embedding is then used to condition the model construction on the task. This is achieved by concatenating the task embedding to every input that is fed to the controller RNN.

   Specifically, in single-task NAS, the controller RNN generates an output at each timestep that determines the distribution over the current set of actions. An action is sampled according to this action distribution and then embedded. The action embedding is then passed back into the RNN as input to the next timestep.

   Here, in multi-task training for MNMS, for multi-task training, the task embedding is now concatenated with the action embedding to form the RNN input, allowing it to condition each action on a specific task (Figure 2).

2. *Off-policy training using multitask replay.*

   Previous works train the NAS controller using the REINFORCE policy gradient algorithm (Zoph & Le, 2017), and more recently, the PPO algorithm (Zoph et al., 2017; Bello et al., 2017).

   Here, we train using off-policy PPO, an actor-critic algorithm in which an actor controller generates sampled models and a critic controller trains on a replay bank of the sampled models and rewards (Schulman et al., 2017). Preliminary experiments with on-policy training found that the controller shows a reduced ability to learn a differentiated model for each task. Specifically, the controller is prone to premature convergence to a single model design that works generally well for all tasks but is not the best model for some of the tasks.

   On-policy sampling is biased toward more recent predictions of the optimal parameter distribution. Our hypothesis is that in multitask training, on-policy sampling can prematurely reduce exploration of better parameters for each individual task, while off-policy training allows the actor controller to continue to explore separate parameter choices for each task, and better learn a differentiated distribution over the parameter search space to maximize expected performance for each.

3. *Per-task baseline and reward distributions normalization.*

   Each task can define a different performance metric to be used as reward. The rewards affect the amplitude of the updates on the controller, so we need to make sure that the distributions of each task rewards are aligned to have same mean and similar variance.

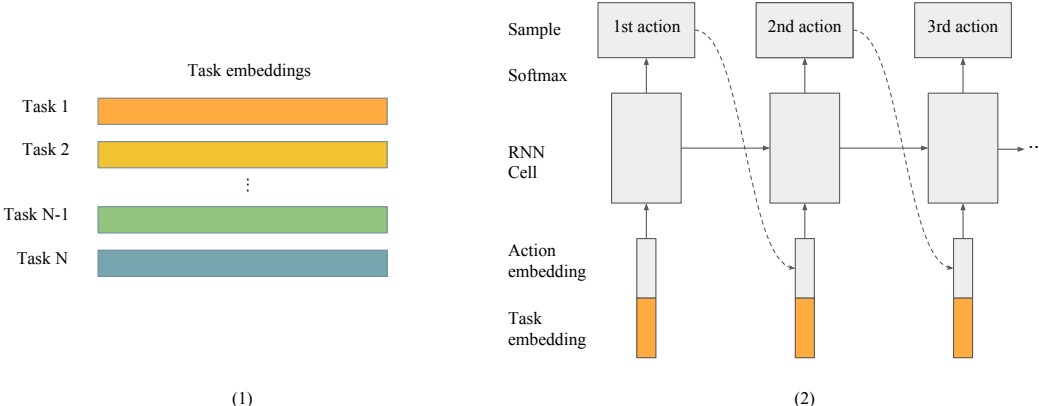

Figure 2: Overview of the multitask controller RNN. (1) A task embedding table is maintained and updated with controller gradients to learn differentiated task embeddings over time. (2) At each iteration of the multitask training, a task is randomly sampled. The task embedding is passed into the controller RNN along with the sampled action embedding at each RNN timestep. The full sequence of output actions defines the child architecture trained on the chosen task.

The mean of each tasks reward distribution is aligned to 0 by scaling the gradients with the advantage instead of the reward. The advantage, $A(a, t)$, of a given model, $a$, applied to a task, $t$, is defined as the difference between the reward, $R(a, t)$, and the expected reward for the given task, $b(t)$:

$$A(a, t) = R(a, t) - b(t)$$

$b(t)$ is often referred to as the baseline. This is a standard RL technique that is usually applied with the aim of increasing training stability. During multitask training, the baseline is conditioned on the sampled task. We keep track of a separate baseline for each task, computed as an exponential moving average of the rewards recorded for each task.

The range of each tasks reward distribution is normalized by dividing the advantage by the baseline:

$$A'(a, t) = \frac{R(a, t) - b(t)}{b(t)}$$

We refer to $A'$ as the normalized advantage. Notice that the division by the baseline does not compromise the convergence criteria, as it can be seen as using a distinct adaptive learning rate for each task.

Using the normalized advantage to scale the gradients instead of the raw reward allows MNMS to use any performance metric as a reward even when training on multiple tasks.

## 3.3 Transfer Learning for Automated Model Search

Using the multitask framework, we can transfer learn pretrained controllers by simply reusing the weights of the pretrained controller and adding a randomly initialized task embedding for each new task. The controller weights and the new task embedding are then updated with standard policy gradient steps.

In our experiments, we also restart the experience replay bank used by the off-policy critic, so that only rewards obtained on the new task are sampled. However, future work could retain and continue to sample from previously seen tasks in order to better retain controller memory of the former tasks.

## 4 EXPERIMENTS AND RESULTS

We apply MNMS to the NLP setting, demonstrating that the framework can be trained simultaneously to design models for two separate text classification tasks. We then transfer learn the MNMS model to two new text classification tasks, and demonstrate that the pre-trained framework achieves significant speedups in model search.

Additional details about the experimental procedures and results follow.

### 4.1 EXPERIMENT SETUP

*Tasks*

For multitask training, we trained the MNMS framework simultaneously on two text classification tasks:

1. Binary sentiment classification on the Stanford Sentiment Treebank (SST) dataset (Socher et al., 2013).
2. Binary Spanish language identification on a dataset consisting of each of the 5,000 highest frequency Wikipedia tokens in English, Spanish, German, and Japanese. The example label is a binary label denoting whether the token is Spanish or not Spanish.

These tasks were chosen specifically for their differences in task complexity, language, and potential for overfitting. This would require a controller capable of true multitask model search to differentiate between the tasks when choosing optimal model parameters for each task.

For transfer learning, we then trained the pre-trained MNMS framework on two new text classification tasks:

1. Binary sentiment classification on the IMDB Large Movie Review dataset, which consists of 50,000 English movie reviews (Maas et al., 2011).
2. Binary sentiment classification on the CorpusCine dataset, which consists of 3,878 Spanish movie reviews from the MuchoCine website (Cruz et al., 2008).

These tasks were chosen so that an effectively transfer learned framework could conceivably leverage knowledge from previous searches. As a baseline to compare search convergence rates, we also trained MNMS models from scratch on the transfer learning tasks.

*Search Space*

For all four tasks, we define a single general search space consisting of 7 common model parameters, with 2-6 discrete parameter choices specified for each (Table 1). A naive grid search over all parameters would therefore need to try 15,360 parameter combinations to search over all possible models. These parameters represent general architectural and training design choices applicable to any text classification task.

Child networks are then constructed as feed-forward neural networks using the sampled parameters. Specifically, for a sampled parameter sequence consisting of word embedding $W$, word embedding trainability $T$, number of neural network layers $N_{layers}$, number of nodes per layer $N_{nodes}$, learning rate $L$, number of training iterations $I$, and $L2$ regularization weight $w$, we construct a feedforward network with $N_{layers}$ RELU-activated layers and $N_{nodes}$ per layer. For each task, the network receives tokens embedded using $W$, where we continue to gradient update the entire word embedding table if $T$ is true. The child model is trained for $I$ iterations using learning rate $L$ and $L2$ regularization weight $w$. All child models end with a final fully-connected softmax layer, and are trained using the Proximal Adagrad optimizer on batches of 100 training examples at each iteration.

*Training Details*

The actor and critic controller RNNs used in off-policy PPO training are 2-layer LSTMs with hidden layer size 50. At each RNN timestep, both action and task embeddings have size 25, resulting in an RNN input of size 50 after concatenation. Both controller and embedding weights are initialized uniformly at random between -0.08 and 0.08.

Table 1: The search space, consisting of six commonly-tuned architectural and training parameters for NLP tasks.

| PARAMETER | PARAMETER CHOICES |
| --- | --- |
| Word embedding tables | {Spanish, German, Japanese, English-small, English-big, English-wiki} |
| Word embedding trainability | {True, False} |
| Number of neural network layers | {1, 2, 3, 5, 10} |
| Number of nodes per hidden layer | {5, 10, 50, 100} |
| Learning Rate | {0.001, 0.01, 0.05, 0.1} |
| Number of training iterations | {5,000, 10,000, 15,000, 20,000} |
| L2 Regularization weight | {0, 0.0001, 0.001, 0.01} |

Table 2: Details of the Word embedding tables.

| LANG./ID | DIMENSIONS | VOCAB SIZE | TRAINING | TOKENS |
| --- | --- | --- | --- | --- |
| Spanish | 128 | 995k | Cont. BOW | 50B |
| German | 128 | 998k | Cont. BOW | 30B |
| Japanese | 128 | 993k | Cont. BOW | 6B |
| English-small | 50 | 982k | Cont. BOW | 7B |
| English-big | 128 | 999k | Cont. BOW | 200B |
| English-wiki | 250 | 1M | Skipgram | 4B |

When training, the controller that receives gradient updates is trained on batches of size 20 with learning rate $5 \cdot 10^{-4}$, and updated for 25 gradient steps before the weights between the two controllers are averaged with Polyak Average weight 0.9. The reward used for updating the controller is the cubed accuracy on a validation set.

## 4.2 SIMULTANEOUS MULTITASK TRAINING RESULTS

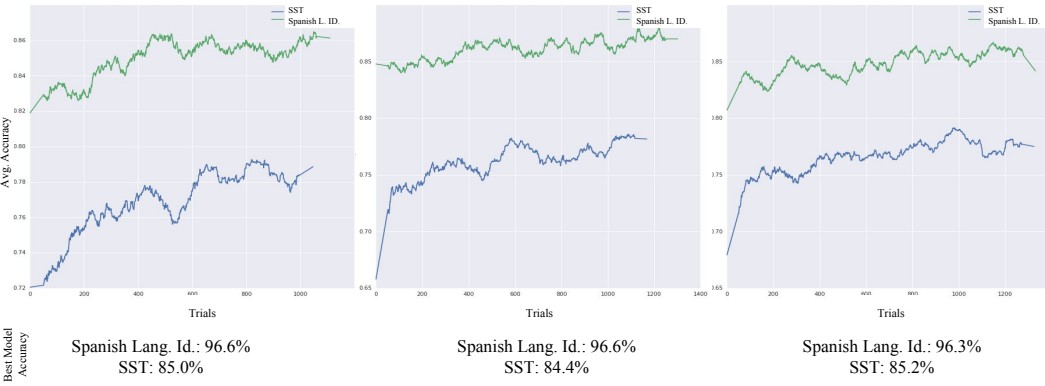

Benchmark hand-tuned VecAvg model with sentence-level annotations, Socher 2013: 80.1%

Figure 3: Smoothed sampled model accuracy curves for multitask NMS when training simultaneously on the Spanish language identification and SST tasks, and the best validation accuracy achieved for each task. Curves shown use Savitzky-Golay filtering with n=101 for clarity. The reference benchmark accuracy by Socher (2013) was achieved on the SST task.

We train n=3 MNMS models simultaneously on the SST and Spanish language identification tasks. Accuracies achieved by the sampled child models over time are shown in Figure 3, as well as the validation accuracy achieved by the best sampled models for each task. In each model, the accuracy

of sampled models improves over time on a per-task basis, even while the tasks clearly have different baseline accuracies.

Additionally, we find that the best discovered model design outperforms the hand-tuned state-of-the-art model within the subset of models that use a similar BOW approach (Socher et al., 2013) The best performance on the task is obtained by more complex architectures that are not within the scope of our search space (Le & Mikolov, 2014).

We also find that the MNMS framework can differentiate between the tasks to choose optimal parameters for each. In Figure 4, we show that MNMS learns differentiated distributions over the parameter search space for the separate tasks. For example, MNMS learns to choose a word embedding pre-trained on Spanish documents for the Spanish language identification task, while choosing word embeddings pre-trained on an English dataset for the Stanford Sentiment Treebank task.

Finally, we find that MNMS learns that for the trivial language identification task, there is no significant difference between continuing to train the word embedding vectors or simply using the fixed, pre-trained word embeddings. For the SST task, which contains longer and more complex examples, the model learns that it must continue training the word embeddings to achieve better performance. Similarly, the search converges to favor higher hidden layer dimensions and more training iterations for the more complex SST task.

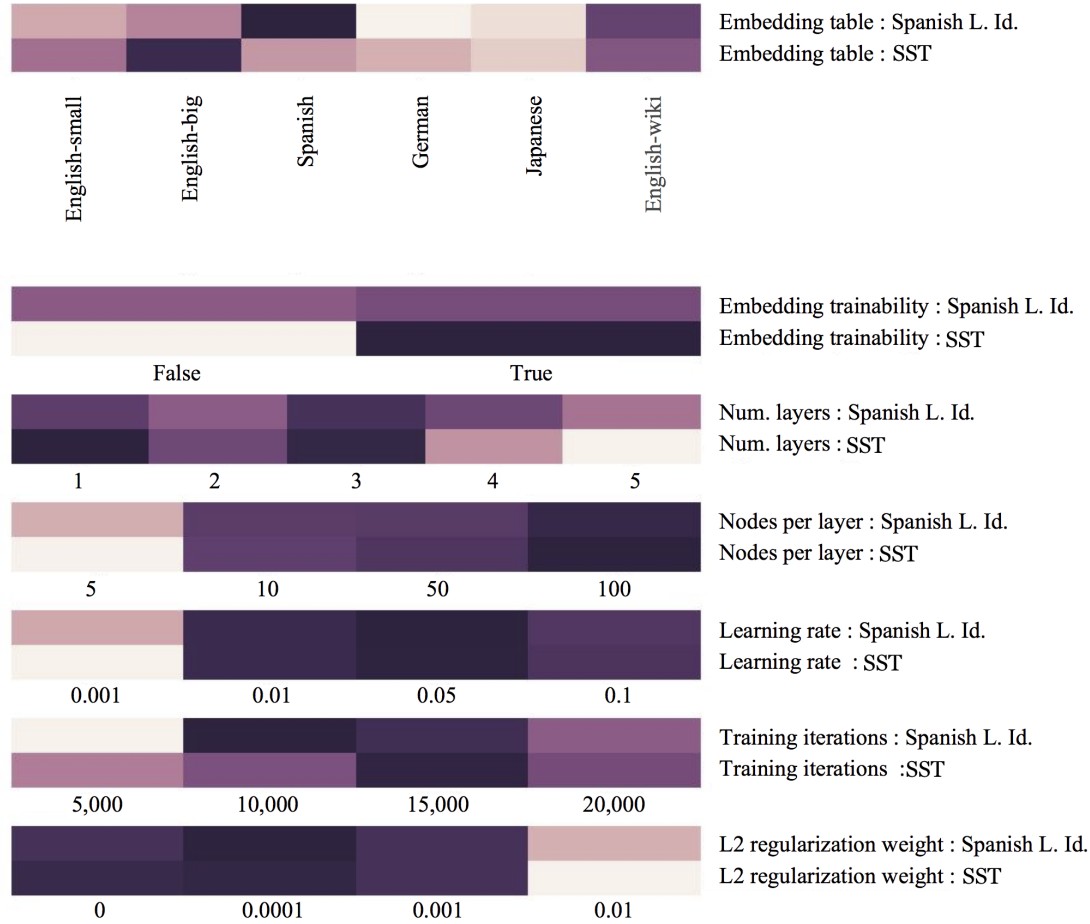

Figure 4: Heatmap showing the learned per-task distributions over the parameter search space from a representative MNMS model.

### 4.3 TRANSFER LEARNING RESULTS

Figure 5 compares the smoothed validation accuracy curves of baseline MNMS models trained from scratch on the IMDB and Corpus Cine tasks with MNMS models pre-trained on SST and Spanish language identification. We observe that transfer learning allows MNMS to start from a better initial location in the parameter search space, train more consistently and stably, and converge much more quickly to finding good parameters for the tasks. Additionally, we find that the best learned models discovered by MNMS perform essentially identically regardless of whether the search is started from scratch or transfer learned from a pre-trained model, demonstrating that the search is not so biased towards pre-training that it converges prematurely to local optima. When compared against other hand-tuned, state-of-the-art benchmarks also using averaged word vector inputs, we find that MNMS discovers models that outperform documented benchmarks on both tasks (Maas et al., 2011; Calvo, 2017).

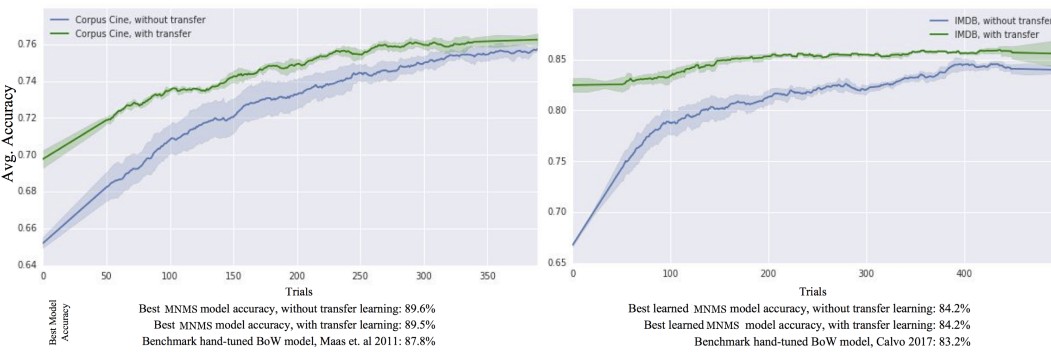

Figure 5: Smoothed sampled model accuracy curves for n=3 MNMS models trained on IMDB and Corpus Cine, comparing models trained from scratch without transfer learning, and models transfer learned after pre-training. Curves smoothed using Savitzky-Golay filtering (n=101) for clarity.

We also find that MNMS learns task embeddings that encode expected relationships between the tasks (Figure 6). For example, we see a strong learned correlation between the IMDB and SST task embeddings, and separately between the Spanish language identification and Corpus Cine task embeddings. While correlation offers one metric to compare these embeddings with intuitive relationships between the tasks, however, future work could attempt to learn more disentangled and interpretable representations. Notably, for example, the Corpus Cine task embeddings are not as strongly correlated with the SST task embeddings, even though both are sentiment analysis tasks. Future work could explore which dimensions within the task embeddings actually differ, and attempt to draw human-interpretable insights that could improve future model designs on similar tasks.

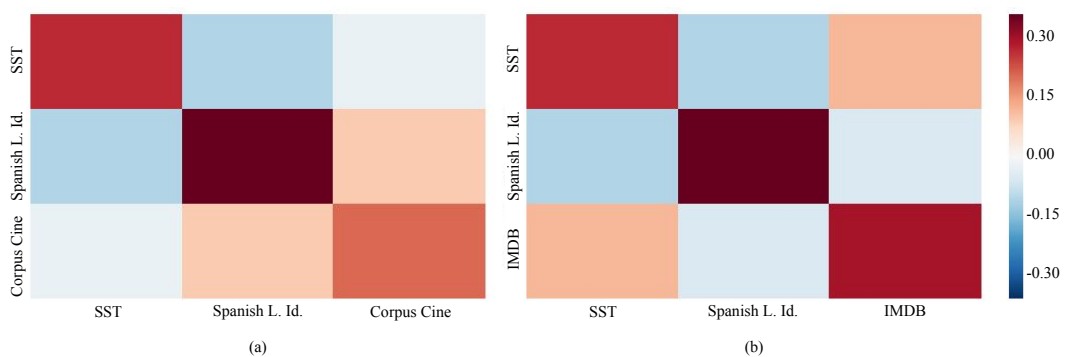

Figure 6: Heatmap showing correlations between learned task embeddings in a pre-trained MNMS controller transfer learned on (a) the Corpus Cine and (b) the IMDB sentiment classification tasks.

## 5 DISCUSSION

*Summary.* Machine learning model design choices do not exist in a vacuum. Human experts design good models by leveraging significant prior knowledge about the intuitive relationships between these model parameters, and the performance obtained by different model designs on similar tasks. Automated model design algorithms, too, can and should learn from the models they have discovered for prior tasks. This paper demonstrates that Multitask Neural Model Search can discover good, differentiated model designs for multiple tasks simultaneously, while learning task embeddings that encode meaningful relationships between tasks. We then show that multitask training provides a good baseline for transfer learning to future tasks, allowing the MNMS framework to start from a better location in the search space and converge more quickly to high-performing designs.

*Limitations and future work.* While the current work demonstrates that the MNMS framework can be used for multitask training and transferable architecture searches, much work remains to determine the scalability of this approach. The results of this study offer several particularly promising avenues for future research. First, studying the effects of additional simultaneous tasks on framework performance is an obvious next step in multitask training. The current framework trains the learned task embeddings by passing them directly into the controller RNN along with the sampled action embeddings. We anticipate that a more complex pre-processing structure, such as a simple encoder-decoder, could better transform these task embeddings to be used by the controller. Additionally, we currently leverage the distributed training structure described by Zoph and Le, which trains multiple sampled child architectures in parallel and asynchronously updates a shared controller parameter server (Zoph & Le, 2017). However, as we continue to scale the MNMS framework for additional simultaneous tasks, future work remains to optimize a parallel training structure and schedule specifically for efficient multitask training.

Experimenting with broader richer hyperparameter search spaces also offers an exciting line of future work. For our current tasks, we defined a search space that encompassed a range of general design choices, including both real-valued parameters (such as learning rates and regularization weights) and higher-level parameters (such as the choice of word embedding table). However, we are actively adapting the controller to sample continuous real-valued parameters, rather than discrete choices from a set of predefined values, which would give the framework much greater flexibility in specifying models. Additionally, we plan to continue expanding the range of modular, higher-level parameter choices in the search space. Allowing the controller to compose these building blocks, rather than more granular design choices, can allow the framework to construct more complex architectures in much less time.

Finally, much work remains to explore cases when transfer learning is and is not effective within RL-based architecture search frameworks such as MNMS. We are particularly interested in studying how transfer learning can be used to design architectures for tasks that were previously considered too resource intensive for standard NAS. For example, Zoph et al. (2017) adapted NAS for the ImageNet classification task by directly modifying the architecture designed for a simpler image classification task. However, pretraining the architecture search framework itself on more computationally feasible tasks, rather than transferring the discovered architectures, would be a significant step towards tackling these difficult search domains.

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
