# OpenReview forum: "Transfer Learning to Learn with Multitask Neural Model Search"
_ICLR.cc/2018/Conference — Reject_

### Official Review · AnonReviewer1 · 2017-11-25
**Nice project, a bit slim on the empirical side**

**Rating:** 5
**Confidence:** 2

**Review:**

The paper proposes an extension of the Neural Architecture Search approach, in which a single RNN controller is trained with RL to select hyperparameters for child networks that must perform different tasks. The architecture includes the notion of a "task embedding", that helps the controller keeping track of similarity between tasks, to facilitate transfer across related tasks.

The paper is very well written, and based on a simple but interesting idea. It also deals with core issues in current machine learning.

On the negative side, there is just one experiment, and it is somewhat limited. In the experiment, the proposed model is trained on two very different tasks (English sentiment analysis and Spanish language detection), and then asked to generalize to another English sentiment analysis task and to a Spanish sentiment analysis task. The models converge faster to high accuracy in the proposed transfer learning setup than when trained one a single task with the same architecture search strategy. Moreover, the task embedding for the new English task is closer to that of the training English task, and the same for the training/test Spanish tasks.

My main concern with the experiment is that the approach is only tested in a setup in which there is a huge difference between two classes of tasks (English vs Spanish), so the model doesn't need to learn very sophisticated task embeddings to group the tasks correctly for transfer. It would be good to see other experiments where there is less of a trivial structure distinguishing tasks, to check if transfer helps.

Also, I find it surprising that the Corpus Cine sentiment task embedding is not correlated at all with the SST sentiment task. If the controller is really learning something interesting about the nature of the tasks, I would have expected a differential effect, such that IMDB is only correlated with SST, but Corpus Cine is correlated to both the Spanish language identification task and SST. Perhaps, this is worth some discussion.

Finally, it's not clear to me why the multitask architecture was used in the experiment even when no multi-task pre-training was conducted: shouldn't the simple neural architecture search method be used in this case?

Minor points:

"diffferentiated": different?

"outputted actions": output actions

"the aim of increase the training stability": the aim of increasing training stability

Insert references for Polyak averaging and Savitzky-Golay filtering.

Figure 3: specify that the Socher 2013 result is for SST

Figure 4: does LSS stand for SST?

I'm confused by Fig. 6: why aren't the diagonal values 100%?

MNMS is referred to as MNAS in Figure 5.

For architecture search, the neuroevolution literature should also be cited (https://www.oreilly.com/ideas/neuroevolution-a-different-kind-of-deep-learning).

---

> ### Author Response · Authors · 2018-01-05
> **Thank you!**
>
> Again, thank you for the thoughtful and detailed review. In addition to our responses above to the other two reviews, to first clarify the nature of the experiments:
>         	1. Section 4.2 describes the results of training MNMS models jointly on the SST and Spanish Language Identification tasks.
>         	2. Section 4.3 uses these multi-task-trained MNMS models as the pre-trained models. The transfer learning results shown are the result of subsequently training the MNMS models initialized to the weights from part 4.2 further on each of the transfer learning tasks (CorpusCine and IMDB). During transfer learning, we initialize a new task embedding vector for the new task that is again trained jointly with the MNMS model. While we transfer learn to a single new task, multi-task pretraining has occurred. Further, after transfer learning, the learned task embedding for the new task can now be directly and meaningfully compared to the existing task embeddings, as in Figure 6.
>
> 1. “My main concern with the experiment is that the approach is only tested in a setup in which there is a huge difference between two classes of tasks (English vs Spanish), so the model doesn't need to learn very sophisticated task embeddings to group the tasks correctly for transfer. It would be good to see other experiments where there is less of a trivial structure distinguishing tasks, to check if transfer helps.”
>         	We specifically chose the two initial multitask tasks to be different enough that a single set of hyperparameters would not be optimal for both. However, as seen in 4.2, there are other significant differences in the parameters learned for each task beyond the English vs Spanish word embeddings.
>         	We have revised the draft to include a further discussion within the conclusion section of the limitations of these experiments and necessary future tasks to demonstrate generalization.
>
> 2. “Also, I find it surprising that the Corpus Cine sentiment task embedding is not correlated at all with the SST sentiment task. If the controller is really learning something interesting about the nature of the tasks, I would have expected a differential effect, such that IMDB is only correlated with SST, but Corpus Cine is correlated to both the Spanish language identification task and SST. Perhaps, this is worth some discussion.”
> This is a good point, and we have updated the discussion to touch on this.
>
> 3. “Finally, it's not clear to me why the multitask architecture was used in the experiment even when no multi-task pre-training was conducted: shouldn't the simple neural architecture search method be used in this case?”
>         	As clarified above, the transfer learning experiments show the results after multi-task pre-training. Let us know if further clarification can be made.
>
> 4. Minor points: thank you for catching these. We have updated the grammatical fixes where we believed appropriate and the figures accordingly, and added a reference to the neuroevolution literature.

---

> > ### Comment · AnonReviewer1 · 2018-01-05
> > **Thanks for reply, missing revision?**
> >
> > Thanks for your reply. Unfortunately, I do not find it as helpful as it could be, because it refers to a revision of the paper that, as far as I can see, you have not uploaded on the OpenReview site. Consequently, you're pointing to arguments and references I cannot access :(
> >
> > Concerning the other differences in 4.2, perhaps some ablation would help to see how much they matter?

---

> > > ### Author Response · Authors · 2018-01-05
> > > **Revision uploaded**
> > >
> > > Apologies - the revision is now uploaded.

---

> > > > ### Comment · AnonReviewer1 · 2018-01-06
> > > > **thanks for the revision**
> > > >
> > > > Thanks for the revision, that clarifies some important points and puts the results in perspective. While I find the general direction of your work very promising, I stand by my initial point of view that more extensive experiments should be added for a long paper in a major conference.

---

### Official Review · AnonReviewer3 · 2017-11-27
**The neural architecture design is important and interesting**

**Rating:** 7
**Confidence:** 3

**Review:**

Summary
This paper extends Neural Architecture Search (NAS) to the multi-task learning problem. A task conditioned model search controller is learned to handle multiple tasks simultaneously. The experiments are conducted on text data sets to evaluate the proposed method.

Pros
1.	The problem of neural architecture design is important and interesting.
2.	The motivation is strong. NAS (Zoph & Le, 2017) needs to train a model for a new task from scratch, which is inefficient. It is reasonable to introduce task embeddings into NAS to obtain a generalization model for multiple tasks.

Cons
1.	Some important technical details are missing, especially for the details regarding task embeddings.
2.	The experiments are not sufficient.

Detailed Comments
1.	The paper does not provide the method of how to obtain task embeddings. In addition, if task embeddings are obtained by an auxiliary network, is it feasible to update task embeddings by updating the weights of this auxiliary network?
2.	The discussion of off-policy training is questionable. There is no experiment to demonstrate the advantage of off-policy training compared to on-policy training.
3.	In order to demonstrate the effectiveness of the idea of multi-task learning and task conditioning in MNMS, some architecture search methods for single-task should be conducted for comparison. For instance, NAS on SST or the Spanish language identification task should be compared.
4.	In order to demonstrate the efficiency of MNMS, running time results of MNMS and NAS should be reported.
5.	In my opinion, the title is not appropriate. The most important contribution of this paper is to search neural models for multiple tasks simultaneously using task conditioning. Only when this target is achieved, is it possible to transfer a pre-trained controller to new tasks with new task embeddings. Therefore, the title should highlight multitask neural model search rather than transfer learning.
6.	In Figure 5, "MNAS" should be "MNMS".

---

> ### Author Response · Authors · 2018-01-05
> **Thank you!**
>
> Thank you for your thoughtful and detailed review! In addition to our response above, to address the other detailed comments within your review:
> 1. “The paper does not provide the method of how to obtain task embeddings. In addition, if task embeddings are obtained by an auxiliary network, is it feasible to update task embeddings by updating the weights of this auxiliary network?”
>         	As described in 3.2, the task embeddings are randomly initialized vectors that are trained jointly with the controller; these embeddings are therefore learned automatically as part of the training process. During transfer learning to a new task, a new, randomly-initialized vector representation is added to the embedding table for the new task, and the task embedding for the new task is again learned automatically during transfer learning.
>         	As with other embedding tables, it is possible to continue updating all of the existing task embeddings along with other network weights during subsequent transfer learning. In this work, the same pre-trained model is separately transfer learned to each of the IMDB and CorpusCine tasks. We therefore do not continue to update the initial pre-training task embeddings here to allow between comparison between the transfer learned tasks.
>
> 2. “The discussion of off-policy training is questionable. There is no experiment to demonstrate the advantage of off-policy training compared to on-policy training.”
>
> 3. “In order to demonstrate the effectiveness of the idea of multi-task learning and task conditioning in MNMS, some architecture search methods for single-task should be conducted for comparison. For instance, NAS on SST or the Spanish language identification task should be compared.”
>         	Figure 5 shows the performance of an MNMS model trained on a single task (Corpus Cine and IMDB) as a baseline for comparison with the transfer learned models.
> We present the task conditioning for simultaneous task training as a stepping stone towards more generalized training for transfer learning to new tasks, rather than as a method for run-time improvements in itself.
>
> 4. “In order to demonstrate the efficiency of MNMS, running time results of MNMS and NAS should be reported.”
>         	MNMS as presented is a direct generalization of NAS, and in the single-task case (as with the single-task, non-pre-trained baselines compared with transfer learning) frameworks are identical. Training with a single, randomly initialized task embedding is equivalent to simply using a standard RNN embedding in the vanilla NAS framework.
>
> 5. “In my opinion, the title is not appropriate. The most important contribution of this paper is to search neural models for multiple tasks simultaneously using task conditioning. Only when this target is achieved, is it possible to transfer a pre-trained controller to new tasks with new task embeddings. Therefore, the title should highlight multitask neural model search rather than transfer learning.”
>         	While we do believe that multitask transfer learning is an important contribution, multitask transfer learning is presented as a stepping stone specifically towards enabling transfer learning. We present multitask training and the concept of task embeddings for task conditioning as a method to enable generalization for automated architecture design that can extend to new tasks.
>
> 6. “In Figure 5, "MNAS" should be "MNMS".”
>         	This has been updated in the revised draft; thank you for catching this!

---

### Official Review · AnonReviewer2 · 2017-11-27
**Initial work on building a framework for finding best performing NN architecture across multiple tasks simultaneously**

**Rating:** 4
**Confidence:** 4

**Review:**

In this paper authors are summarizing their work on building a framework for automated neural network (NN) construction across multiple tasks simultaneously.

They present initial results on the performance of their framework called Multitask Neural Model Search (MNMS) controller. The idea behind building such a framework is motivated by the successes of recently proposed reinforcement based approaches for finding the best NN architecture across the space of all possible architectures. Authors cite the Neural Architecture Search (NAS) framework as an example of such a framework that yields better results compared to NN architectures configured by humans.

Overall I think that the idea is interesting and the work presented in this paper is very promising. Given the depth of the empirical analysis presented the work still feels that it’s in its early stages. In its current state and format the major issue with this work is the lack of more in-depth performance analysis which would help the reader draw more solid conclusions about the generalization of the approach.

Authors use two text classification tasks from the NLP domain to showcase the benefits of their proposed architecture. It would be good if they could expand and analyze how well does their framework generalizes across other non-binary tasks, tasks in other domains and different NNs. This is especially the case for the transfer learning task.

In the NAS overview section, readers would benefit more if authors spend more time in outlining the RL detail used in the original NAS framework instead of Figure 1 which looks like a space filler.

Across the two NLP tasks authors show that MNMS models trained simultaneously give better performance than hand tuned architectures. In addition, on the transfer learning evaluation approach they showcase the benefit of using the proposed framework in terms of the initially retrieved architecture and the number of iterations required to obtain the best performing one.
For better clarity figures 3 and 5 should be made bigger.
What is LSS in figure 4?

---

> ### Author Response · Authors · 2018-01-05
> **Thank you**
>
> Thank you for the thoughtful and detailed review. We are actively continuing to evaluate the MNMS framework on additional search spaces, non-binary tasks, and tasks outside of the NLP domain. However, we believe that the presented experiments are sufficient to demonstrate two important contributions of the proposed generalized framework, each of which addresses key concerns about previous RL-based automated NN design frameworks such as NAS:
> 1. Simultaneous task training is possible.
>         	The ability to handle multitask training of any kind addresses key issues regarding
>         	the generalizability and feasibility of RL-based automated search frameworks. In
>         	particular, learning task representations that allow a single controller model to
>         	differentiate between tasks is necessary for any kind of task generalization using
>         	this form of meta-learning framework.
>         	However, especially in RL-based environments, prior work has
>         	demonstrated that handling multitask learning is empirically challenging,
>         	even on relatively simple tasks; as the two papers cited in the related work section
>         	also discuss, multitask RL training even across two tasks often causes negative
>         	interference between the tasks, including cases where gradients from one task
>         	completely dominate the other. Therefore, it is not obvious that a
>         	NAS-like framework should be able to handle multitask training, even on
> relatively simple domains, rather than simply collapsing to single, undifferentiated
> parameter choices that are suboptimal for each task. Indeed, as we describe in 3.2,
> multitask replay is necessary even in this relatively simple domain to ensure adequate
> differentiation.
>         	Therefore, while preliminary, we believe that it is important to show that an RL-
>         	based metalearning framework can indeed discover differentiated architectures for
>         	two tasks, which were specifically chosen so that no single, optimal parameter
>         	solution existed. Further, the ability to automatically learn vector
>  task representations sufficient to encode this differentiation during training, even
> in this relatively simple task domain, offers a necessary step towards further work
> in simultaneous multitask training across more challenging tasks in the future.
> 2. Pre-training NAS-like frameworks for future transfer learning to new tasks is possible, and speeds up convergence.
>         	A primary criticism of RL-based metalearning architectures, such is NAS, is that these methods are extremely time and computationally intensive, rendering them infeasible without computational resources that are not accessible to many researchers. Therefore, the possibility of using pre-trained models for transfer learning to any new task to reduce search time is a necessary step towards making this approach broadly feasible, both for additional research and more challenging tasks.
>         	However, as with multitask training, it is not obvious that this would be possible, even when designing architectures for relatively simple task. Again, especially in RL models, attempting to transfer learn could lead to either 1. premature convergence to suboptimal parameters biased by the pre-training, or 2. no convergence speedup, or even additional convergence time, in which the controller first unlearns its pre-training and then learns the new task. Figure 5 in the results shows that transfer learning in this domain allows the controller to 1. start from a better place in the search space, indicating that NAS-like frameworks can learn knowledge that generalizes to another task, and 2. converge more quickly overall, indicating that this transfer learning can speed up convergence and therefore opening transfer learning as a grounds for further research.
> Therefore, while these empirical results are preliminary, we believe that demonstrating that both of these points are possible are important, nonobvious generalizations on the NAS architecture that offer routes for future study.

---

> > ### Author Response · Authors · 2018-01-05
> > **Thank you**
> >
> > To further address the points made in this review specifically:
> >
> > 1. “Given the depth of the empirical analysis presented the work still feels that it’s in its early stages. In its current state and format the major issue with this work is the lack of more in-depth performance analysis which would help the reader draw more solid conclusions about the generalization of the approach.”
> >         	As discussed above, this paper was intended to propose a generalized framework and demonstrate that both multitask training and transfer learning are possible, within these proof-of-concept domains. However, please let us know if there are suggestions for specific further analyses about the current experiments.
> >
> > 2. “It would be good if they could expand and analyze how well does their framework generalizes across other non-binary tasks, tasks in other domains and different NNs. This is especially the case for the transfer learning task.”
> >         	We have updated the revised draft conclusion to include a more detailed discussion of the limitations of this current study, and to include further discussion of ongoing work and future work to evaluate the framework on additional tasks and task sets, based on this feedback.
> > 3. “In the NAS overview section, readers would benefit more if authors spend more time in outlining the RL detail used in the original NAS framework instead of Figure 1 which looks like a space filler.”
> > 	While this section was intended as a minimal overview of the original NAS framework (with the understanding that readers could reference the original works for greater detail), we have updated the revised draft to include some additional details, and reduced the size of figure 1.
> >        .
> >
> > 4 and 5:  “For better clarity figures 3 and 5 should be made bigger.” and “What is LSS in figure 4?”
> > 	The revised draft corrects the typo (LSS is now SST) as well as a typo in Figure 3.
> >         	We have updated the revised draft to enlarge figures 3 and 5, and correct the typo (LSS is now SST.) Thank you!

---

### Decision · Program_Chairs · 2018-01-29
**ICLR 2018 Conference Acceptance Decision**

**Decision:**

Reject

**Comment:**

This paper presents a sensible, but somewhat incremental, generalization of neural architecture search.  However, the experiments are only done in a single artificial setting (albeit composed of real, large-scale subtasks).  It's also not clear that such an expensive meta-learning based approach is even necessary, compared to more traditional approaches.

If this paper was less about proposing a single new extension, and more about putting that extension in a larger context, (either conceptually or experimentally), it would be above the bar.